# Gene Expression of Neurogenesis Related to Exercise Intensity in a Cerebral Infarction Rat Model

**DOI:** 10.3390/ijms25168997

**Published:** 2024-08-19

**Authors:** Min-Keun Song, Hyun-Seok Jo, Eun-Jong Kim, Jung-Kook Kim, Sam-Gyu Lee

**Affiliations:** Department of Physical & Rehabilitation Medicine, Chonnam National University Medical School, #160, Baekseo-ro, Dong-gu, Gwangju 61469, Republic of Korea; drsongmk@jnu.ac.kr (M.-K.S.);

**Keywords:** cerebral infarction, aerobic exercise, neurogenesis, neurotrophin, gene expression, functional recovery

## Abstract

Regular exercise improves several functions, including cognition, in patients with stroke. However, the effect of regular exercise on neurogenesis related to cognition remains doubtful. We investigated the most effective exercise intensity for functional recovery after stroke using RNA sequencing following regular treadmill exercise. Photothrombotic cerebral infarction was conducted for 10-week-old male Sprague-Dawley rats (*n* = 36). A Morris water maze (MWM) test was performed before a regular treadmill exercise program (5 days/week, 4 weeks). Rats were randomly divided into four groups: group A (no exercise); group B (low intensity, maximal velocity 18 m/min); group C (moderate intensity, maximal velocity 24 m/min) and group D (high intensity, maximal velocity 30 m/min). After 4 weeks, another MWM test was performed, and all rats were sacrificed. RNA sequencing was performed with ipsilesional hippocampal tissue. On the day after cerebral infarction, no differences in escape latency and velocity were observed among the groups. At 4 weeks after cerebral infarction, the escape latencies in groups B, C, and D were shorter than in group A. The escape latencies in groups B and C were shorter than in group D. The velocity in groups A, B, and C was faster than in group D. Thirty gene symbols related to neurogenesis were detected (*p* < 0.05, fold change > 1.0, average normalized read count > four times). In the neurotrophin-signaling pathway, the *CHK* gene was upregulated, and the *NF-κB* gene was downregulated in the low-intensity group. The *CHK* and *NF-κB* genes were both downregulated in the moderate-intensity group. The *Raf* and *IRAK* genes were downregulated in the high-intensity group. Western blot analysis showed that *NF-κB* expression was lowest in the moderate-intensity group, whereas *CHK* and *Raf* were elevated, and *IRAK* was decreased in the high-intensity group. Moderate-intensity exercise may contribute to neuroplasticity. Variation in the expression of neurotrophins in neurogenesis according to exercise intensity may reveal the mechanism of neuroplasticity. Thus, *NF-κB* is the key neurotrophin for neurogenesis related to exercise intensity.

## 1. Introduction

Post-stroke cognitive impairment (PSCI), which occurs in up to 60% of stroke survivors and ranges from mild to severe [1], is associated with several adverse outcomes contributing to a lower quality of life [2]. PSCI may also complicate effective and timely treatment [3].

Regular exercise improves cognitive function in patients with stroke [4]. Activity-regulated structural and functional changes in hippocampal cells may modulate neuroplasticity [5]. Exercise can promote increased levels of neurotrophic factors, change the levels of various cytokines, and alter microglial function in the brains of patients with neurodegenerative diseases [6]. Long-term exercise training in rats may enhance memory by regulating brain-derived neurotrophic factor (BDNF) expression and microglial activation [7,8].

Exercise therapy for neurorehabilitation during functional recovery plays a key role in patients with stroke [9]. However, such patients may have difficulty performing exercise properly because mobility and endurance may be restricted by neurologic deficits. An effective therapeutic exercise program should be precisely prescribed for a favorable outcome following a stroke.

Nonetheless, the effective exercise intensity for neurogenesis related to cognitive function remains unclear. A previous study revealed that swimming exercise of moderate-duration intensity is an effective exercise in a cerebral infarction rat model via neuroplasticity [10]. Several studies of treadmill exercise showed that prolonged low-intensity exercise was more effective than high-intensity exercise [11,12]. Other research revealed that high-intensity exercise may inhibit neurogenesis compared with low- to moderate-intensity exercise [13,14], whereas a systematic review reported that acute high-intensity interval training may enhance executive function in patients with stroke [15]. One recent study reported that varying intensities of treadmill exercise improved working memory in rats [16]. Thus, the optimal treadmill exercise protocol for neurogenesis related to cognitive function remains uncertain.

The aim of the present study was to determine the therapeutic exercise intensity that most effectively promotes cognitive functional recovery in stroke using RNA sequencing following regular treadmill exercise.

## 2. Results

### 2.1. Morris Water Maze Test of Neurobehavioral Function

The escape latency on the first day after cerebral infarction was 100.97 ± 36.28 s in group A, 97.67 ± 33.34 s in group B, 115.51 ± 28.23 s in group C, and 115.04 ± 77.44 s in group D. There was no significant difference among groups (*p* = 0.798). After 4 weeks, the escape latencies were 189.90 ± 74.57, 96.89 ± 64.87, 73.51 ± 29.49, and 134.39 ± 47.19 s in groups A–D, respectively. The follow-up escape latencies in groups B, C, and D were significantly shorter than those in group A, and latencies in groups B and C were shorter than those in group D (*p* = 0.001). Repeated-measures analysis of variance (ANOVA) with the Duncan post hoc test showed a significant time × group interaction effect (F = 5.851, *p* = 0.003). However, no significant effect was shown for time (F = 2.097, *p* = 0.157) (Figure 1).

The velocities on the first day after cerebral infarction were 31.37 ± 9.30 cm/s in group A, 27.55 ± 6.22 cm/s in group B, 27.50 ± 8.53 cm/s in group C, and 24.93 ± 7.49 cm/s in group D. There was no significant difference among the groups (*p* = 0.406). After 4 weeks, the velocities were 24.35 ± 2.77, 27.79 ± 4.42, 27.81 ± 3.69, and 20.71 ± 3.71 cm/s in groups A–D. Respectively, the Duncan post hoc correction revealed faster velocities in groups B and C compared with groups A and D after 4 weeks of exercise (*p* = 0.002). However, repeated-measures ANOVA with the Duncan post hoc test showed no significant effects for time and time × group interaction (F = 2.375, *p* = 0.133; F = 0.334, *p* = 0.567) (Figure 2).

### 2.2. RNA Sequencing Data

QuantSeq RNA analysis revealed 17,048 gene symbols. We identified several symbols showing significant fold changes in each exercise group compared with the control group. Thirty gene symbols related to neurogenesis (Iqgap1, Lrrc7, Coq7, Myef2, Mapk3, Vhl, Fuom, Pex5, Il33, Kdm4c, Ldb1, Heyl, Baiap2, Kdm1a, Mapkapk5, Epha2, Enah, Snw1, Nefm, Hsp90aa1, Nr2f6, Btbd6, Bloc1s5, Fbxo7, Cdh11, Trappc4, Cacna1a, Mapk6, Vcan, and Dag1) were detected (*p* < 0.05, fold change > 1.0, average normalized read count [RC] > 4). To use the clustering heatmap, the patterns of gene expression in the low- and moderate-intensity treadmill exercise groups were similar and different from the control and high-intensity treadmill exercise groups (Figure 3).

### 2.3. KEGG Mapper Tool Analysis

We subjected the gene symbols with greater expression in the experimental groups than in the control group (*p* < 0.05, fold change > 1.0, average normalized RC > 4) to the Kyoto Encyclopedia of Genes and Genomes (KEGG) mapper tool analysis. We found changes in gene expression of components of the neurotrophin-signaling pathway after low-intensity treadmill exercise: the *CHK* gene was upregulated, and the *NF-κB* gene was downregulated (Figure 4). After moderate-intensity treadmill exercise, both the *CHK* and *NF-κB* genes were downregulated (Figure 5). In the high-intensity treadmill exercise group, gene expression was significantly different compared with the low- and moderate-intensity exercise groups, with downregulation of the *Raf* and *IRAK* genes (Figure 6).

### 2.4. Western Blot Analyses

Western blot analyses with antibodies to *NF-kB*, *CHK*, *Raf*, and *IRAK* were conducted for the ipsilesional hippocampal tissues 4 weeks following cerebral infarction. The reactivities of *NF-kB* protein were 0.461 ± 0.023 μg protein in group A, 0.518 ± 0.468 μg protein in group B, 0.375 ± 0.037 μg protein in group C, and 0.456 ± 0.079 μg protein in group D. The expression of the *NF-kB* protein in group C was weaker than in groups A, B, and D (*p* = 0.001) (Figure 7).

The reactivities of the *CHK* protein were 0.453 ± 0.005 μg protein in group A, 0.472 ± 0.003 μg protein in group B, 0.484 ± 0.016 μg protein in group C, and 0.588 ± 0.005 μg protein in group D. The expression of the *CHK* protein was stronger in group D than in groups A, B, and C (*p* = 0.05) (Figure 8).

The reactivities of the *Raf* protein were 0.797 ± 0.016 μg protein in group A, 0.906 ± 0.014 μg protein in group B, 0.973 ± 0.038 μg protein in group C, and 1.059 ± 0.018 μg protein in group D. The expression of the *Raf* protein was stronger in group D than in groups A, B, and C (*p* = 0.05) (Figure 9).

The reactivities of *IRAK* protein were 0.252 ± 0.012 μg protein in group A, 0.247 ± 0.011 μg protein in group B, 0.236 ± 0.010 μg protein in group C, and 0.227 ± 0.014 μg protein in group D. The expression of the *IRAK* protein was weaker in group D than in group A (*p* = 0.05) (Figure 10).

## 3. Discussion

Stroke is a neurologic disorder associated with cerebrovascular events, and it often causes functional deficits. Maximal functional recovery is the primary goal for patients with stroke. Therapeutic approaches to functional recovery have been applied in such patients in combination with proper neurorehabilitation management and medical therapy.

In the neurorehabilitation of patients with stroke, the main strategy for neurobehavioral functional recovery is based on neuroplasticity, which is the ability of the brain to transfer neural connections to a different brain area. Neuroplasticity serves to optimize neural connectivity after brain tissue damage [17]. However, neuroplasticity may not be directly correlated with neurobehavioral functional recovery. Exercise intensity may enhance or decrease neurobehavioral functional recovery in stroke neurorehabilitation. Therefore, flexible and appropriate exercise intensity is important for patients with stroke to attain optimal functional recovery based on neuroplasticity.

Neuroplasticity is related to three mechanisms: neurogenesis, angiogenesis, and synaptogenesis. In particular, neurogenesis involves several neurotrophic pathways that promote the proliferation, differentiation, repair, and survival of neural tissues in the brain [18]. Among several neurotrophins that regulate the neurotrophic pathways related to neurogenesis, the best known are BDNF, nerve growth factor (NGF), neurotrophin-3, vascular endothelial growth factor, insulin-like growth factor-1, and erythropoietin in the adult hippocampus [19]. Still, there are many unknown neurotrophic pathways for neurogenesis by which neurotrophins may modulate neuroplasticity in each area of the brain.

Csk homologous kinase (*CHK*) was downregulated in the moderate-intensity group and upregulated in the low-intensity group, compared with the control group. *CHK* may modulate the terminal differentiation of oligodendrocytes and neurons in the central nervous system. *CHK* is involved in the neurite outgrowth of PC12 cells in response to NGF through its participation in TrkA-signaling [20]. However, overexpression of *CHK* in hippocampal neurons facilitates aggregation of cell bodies, axon fasciculation, and changes in neuronal morphology [21]. The level of *CHK* expression may enhance or inhibit neurogenesis.

By contrast, high-intensity treadmill exercise showed different gene expression patterns compared with low- and moderate-intensity exercise. The *Raf* and *IRAK* genes were downregulated with high-intensity treadmill exercise. *IRAK-1* and *IRAK-4* are recruited to the receptor complex [22], and lack of *IRAK-1* expression in neurons may induce recalcitrancy of *NF-κB* activation during Toll-like receptor (TLR) pathway signaling [23]. Further, *Ras/Raf/ERK* signaling participates in the neuronal apoptosis observed in the hippocampus in early post-subarachnoid hemorrhage brain injury [24]. The downregulation of *Raf* and *IRAK* may have inhibited neurogenesis by interrupting neuronal apoptosis in the present research.

The Western blot analysis showed different results: *CHK* and *Raf* expression were elevated, but *IRAK* expression was decreased in high-intensity treadmill exercise. Thus, the regulation of *CHK*, *Raf* and *IRAK* would not be the main pathway of neurogenesis related to exercise intensity if these may affect the process of neurogenesis.

In this study, the regulation of *NF-ĸB* was the most important in these neurotrophic pathways. *NF-κB* in the central nervous system plays a role in neuroplasticity and neurocognitive function. Previous studies showed that activation of *NF-ĸB* through p75NTh may promote the migration of extracellular matrix proteins during nerve regeneration in Schwann cells [25]. *NF-κB* can be activated by BDNF and NGF and may facilitate synaptogenesis by glutamate [26]. Recent studies have reported that NF-κB is a key neurotrophin for learning and memory in mice [27,28]. NF-κB may regulate cognitive functions such as learning and memory by modulating synaptogenesis [29,30,31,32,33] and by regulating dendrite growth [34]. *NF-κB* target genes may be essential for neurogenesis by modulating neurotransmitters, cytokines, and kinases [31,35]. However, we found that the *NF-κB* gene was downregulated in the low- and moderate-intensity exercise groups compared with the control group. Also, expression of the NF-kB protein in the low- and moderate-intensity exercise groups showed the greatest protein decrease relative to the other groups.

Some studies have shown that overexpression of *NF-κB* in a transgenic mouse combined with massive expression of ncaIKK-2 may cause the destruction of granule cells and stimulate astrocytosis [36]. Further, the presence of the p50 subunit of *NF-κB* may affect brain tissue [37]. In p50−/−mice, half of the newborn neurons survived in the dentate gyrus, although spatial short-term memory was impaired [38]. The function of *NF-κB* remains unclear, and further research is needed to determine the effects of the subunits of *NF-κB* in neurogenesis.

*NF-κB* caused excitotoxic damage in the hippocampus in a middle cerebral artery occlusion model via an anti-inflammatory cytokine [37]. Inhibiting *NF-κB* could suppress neuroinflammation in peri-infarcted areas of the brain and protect neuronal cells [39,40]. In the present research, the brain may still have been in the inflammatory period, and long-term exercise may have provided an anti-inflammatory effect in the cerebral infarction rat model after stabilizing neurogenesis in the brain. Therefore, moderate-intensity exercise training may be sufficient to suppress the expression of the *NF-κB* gene as an inflammation-related gene.

In the behavioral test, we found that low- and moderate-intensity exercise had a greater effect than high-intensity treadmill exercise on neurogenesis in the cerebral infarction rat model. The difference in functional recovery would depend on the level of gene expression in each intensity condition, and *NF-κB* is the key neurotrophin for neurogenesis related to exercise intensity. Therefore, prescribing an exercise program of the proper intensity could play a pivotal role in maximal functional recovery during a stroke neurorehabilitation program.

There are some limitations in this study. First, it does not reflect aging and comorbidities as the most important risk factors for stroke. However, in this study, 14-week-old rats may be considered as aged rats after a 4-week exercise program. Previous studies reported that aged rats could have limited recovery of neurologic function because of the lack of neuroprotective factors, immune imbalance, and comorbidities related to stroke risk factors such as obesity and metabolic syndrome [41,42,43]. Further studies are needed to consider aging and comorbidities in stroke experiments. Second, this study showed gene expression with RNA sequencing analyses and performed Western blot analyses for the quantification of proteins related to gene expression. Several studies have used quantitative methods, such as Western blot analyses after RNA sequencing [44,45,46]. However, there are several direct analyses, such as immunofluorescence, enzyme-linked immunosorbent assay (ELISA) or reverse transcription-polymerase chain reaction (RT-PCR), to confirm the mechanism. Therefore, it would have been better to quantify gene expression with immunofluorescence, ELISA, or RT-PCR. Further studies are needed to use direct analysis methods for standardizing the biomechanism of neurogenesis after various exercise intensities.

## 4. Materials and Methods

### 4.1. Experimental Subjects and Ethics Approval

All experimental protocols were approved by the Institutional Animal Care and Use Committee (IACUC) of Chonnam National University (approval number: CNU IACUC-H 02018-2). Thirty-six 10-week-old male Sprague-Dawley (SD) rats (Samtako; Osan, Republic of Korea) were used. All experimental rats were housed following the protocol of the Animal Care Laboratory at Chonnam National University.

### 4.2. Methods

#### 4.2.1. Photothrombotic Cerebral Infarction Rat Model

We induced photothrombotic cerebral infarction in rats using Watson’s method [47]. Every SD rat was anesthetized with 5% isoflurane and maintained with 2% isoflurane in a 70% nitrous oxide and 30% oxygen mixture during the surgical procedure. The operation was performed on a homeothermic plate (Harvard Apparatus; Holliston, MA, USA) to maintain body temperature at 37.5 ± 0.5 °C. Each rat was placed in a stereotactic frame (Stoelting; Wood Dale, IL, USA) in a prone position. After an incision and exposure of the scalp, we injected Rose Bengal dye (50 mg/kg; Sigma-Aldrich Co., St. Louis, MO, USA) into the left femoral vein, and light exposure (3300 K, 150 W; KL 1500 LCD; SCHOTT AG, Mainz, Germany) was applied to the mixed zone of the left M1 and S1, known as the primary sensorimotor cortex, for 10 min.

#### 4.2.2. Regular Treadmill Exercise Program

All rats were randomly assigned to four groups: group A (no treadmill exercise; *n* = 9); group B (regular treadmill exercise with maximal velocity of 18 m/min, 5 days a week for 4 weeks; *n* = 9); group C (regular treadmill exercise with maximal velocity of 24 m/min, 5 days a week for 4 weeks; *n* = 9); and group D (regular treadmill exercise with maximal velocity of 30 m/min, 5 days a week for 4 weeks; *n* = 9). Exercise intensity was decided by Bedford’s theory as 50%, 65%, or 80% of rat maximal oxygen uptake [48]. An electric treadmill machine (Columbus Instruments; Columbus, OH, USA) was used for the exercise. The apparatus consisted of a 3-lane animal exerciser using single-belt construction with dividing walls suspended over the tread surface. The overall dimensions of the treadmill were 80.0 × 50.0 × 50.8 cm (width × depth × height), and the dimensions of each exercise lane were 56 × 12 × 13 cm (width × depth × height). All rats were placed on a moving belt facing away from the electrified grid (stimulus intensity 1.0 mA), and they ran in the direction opposite to the belt movement to acclimate to the treadmill exercise at an inclination of 0°.

#### 4.2.3. Neurobehavioral Test: The Morris Water Maze

Spatial learning memory was measured using the Morris water maze (MWM) test based on the method described by Morris [49]. The MWM test was conducted in a circular metal pool (diameter 184 cm, height 60 cm) filled with water and maintained at 20–25 °C. The pool was divided into four approximately equal quadrants, one of which was allocated as the target quadrant. Visual cues were placed at the perimeter of each quadrant. A circular escape platform (diameter 10 cm, height 38 cm) was placed in the center of the target quadrant. The water level in the pool was adjusted so that the platform was submerged 1 cm below the surface of the water.

The animals were placed in the water maze facing the maze wall at random entry points, which were distributed equally around the maze perimeter. After finding the platform, the rats were allowed to remain there for 10 s before the next trial. Any rat that did not find the hidden platform within 120 s was guided to the platform and allowed to stay there for 15 s. Then, the rat was removed from the pool, dried, and placed back in its holding bin for a period of 5 min, after which the second trial was conducted. All groups were acclimated to the abovementioned pre-training process for 3 consecutive days before induction of photothrombotic cerebral infarction. After the treadmill exercise program, the follow-up MWM test was performed. In this trial, rats had to find the platform within 300 s. The time to reach the platform (escape latency) and the length of the swim path were recorded automatically by a video tracking system (Ethovision Color-Pro^®^; Noldus, Wageningen, The Netherlands). Velocity was calculated by dividing the length of the swim path by the escape latency.

#### 4.2.4. RNA Sequencing Analysis

RNA sequencing analysis followed Kim’s method [50]. RNA sequencing was performed with ipsilesional hippocampal tissue. Total RNA was isolated with TRIzol™ reagent (Thermo Fisher Scientific; Waltham, MA, USA). A NanoDrop^TM^ 2000 spectrophotometer (Thermo Fisher Scientific; Waltham, MA, USA) was used for RNA quantification. QuantSeq 3′ mRNA-Seq Library Prep Kit (Lexogen, Inc.; Vienna, Austria) was used for library preparation. High-throughput sequencing was performed with NextSeq 500 (Illumina, Inc.; San Diego, CA, USA).

QuantSeq 3′ mRNA-Seq reads were aligned using Bowtie 2 [51]. Expressed genes were determined on counts with Bedtools [52]. The RC data were measured by edgeR within R using a Bioconductor [53]. DAVID (https://david.ncifcrf.gov/, accessed on 19 December 2020) and Medline (http://www.ncbi.nlm.nih.gov/, accessed on 19 December 2020) databases were used for gene classification. A clustering heatmap was developed using MultiExperiment Viewer version 4.9.0 (http://mev.tm4.org, accessed on 19 December 2020) based on log_2_ values for the average of normalized data.

The neurotrophin-signaling pathway was elicited using the KEGG Mapper Search and Color Pathway tool We identified the gene symbols with greater expression in the experimental groups than in the control group (*p* < 0.05, fold change > 1.0, average normalized RC > 4). The identified genes were entered on the website, and we clicked ‘included aliases’ and ‘uncolored diagram’. Then, the related pathways were identified [54]. We chose the neurotrophin-signaling pathway listed for the various pathways in each group.

#### 4.2.5. Western Blot Analysis

Hippocampal tissues were homogenized with 300 μL of a lysis buffer. The tissue was immediately put on ice and stored at –80 °C. After centrifugation at 13,000 rpm for 10 min at 4.0 °C, the supernatant liquid was collected (Centrifuge 5424; Eppendorf, Germany). For the detection of the *NF-κB*, *CHK*, *Raf*, and *IRAK* proteins, 20 μg of each sample was separated using 10% sodium dodecyl sulfate-polyacrylamide gel electrophoresis. The gels were transferred to a polyvinylidene difluoride membrane at 300 mA for 1 h. The membranes were stained with Ponceau Red to confirm equal protein loading. Immunodetection of each protein was done by blocking the membrane with 5% nonfat milk for 1 h. Tris-buffered saline, including 0.1% Tween-20 (TBS-T) for 1 h at room temperature. The membranes were rinsed with TBS-T four times within 10 min. Then, the membranes were incubated with the primary antibodies against NF-κB (rabbit polyclonal antibody 1:1000; Thermo Fisher Scientific), CHK (rabbit polyclonal antibody 1:1000; Abcam, Cambridge, UK), *Raf* (rabbit polyclonal antibody 1:1000; Abcam), and *IRAK* (rabbit polyclonal antibody 1:1000; Abcam), overnight at 4.0 °C in TBS-T. Excess antibodies were removed by washing with TBS-T three times over 5 min and for a series of four times each. The membranes were incubated with a horseradish peroxidase-conjugated goat anti-rabbit immunoglobulin G (1:1000 dilution; Thermo Fisher Scientific, US) secondary antibody blocking solution for 1 h. The membranes were rinsed with TBS-T three times within 5 min and for a series of three times each. Immunoreactive bands were visualized by Enhanced Chemiluminescence Plus using Immobilon Western Chemiluminet substrate (Millipore; Burlington, MA, USA). After the protein bands were photographed in grayscale at a resolution of 600 dpi, the bands were quantified using ImageJ (National Institutes of Health; Washington, DC, USA).

#### 4.2.6. Timing of the Experiments

Figure 11 shows the timing of the experiments in this study. Photothrombotic cerebral infarction was conducted for 10-week-old male SD rats (*n* = 36). All rats were acclimated to the abovementioned pre-training process for 3 consecutive days before photothrombotic cerebral infarction. The MWM test was performed before and after a regular treadmill exercise program (5 days/week, 4 weeks). The follow-up MWM test was performed after the treadmill exercise program; all rats were sacrificed after the MWM test to obtain the brain tissues.

#### 4.2.7. Statistical Analyses

Statistical analyses were performed using SPSS version 23.0 (IBM Corp.; North Castle, NY, USA). A repeated-measures analysis of variance with a post hoc analysis was used for the behavioral test. The Chi-squared test was performed for western blot analyses. Data are shown as mean ± standard deviation. Differences were considered statistically significant when the *p*-value was < 0.05.

## 5. Conclusions

Neurobehavioral functional recovery may have been affected by exercise intensity in a cerebral infarction rat model. Moderate-intensity exercise may contribute to neuroplasticity. Variation in the expression of neurotrophins in neurogenesis according to exercise intensity may reveal the mechanism of neuroplasticity, and in our study, *NF-κB* was the key neurotrophin involved in neurogenesis related to exercise intensity. Overall, this study demonstrated that an exercise program of the proper intensity plays a major role in assuring favorable functional recovery, and especially for the recovery of cognitive function, in stroke rehabilitation.

## Figures and Tables

**Figure 1 ijms-25-08997-f001:**
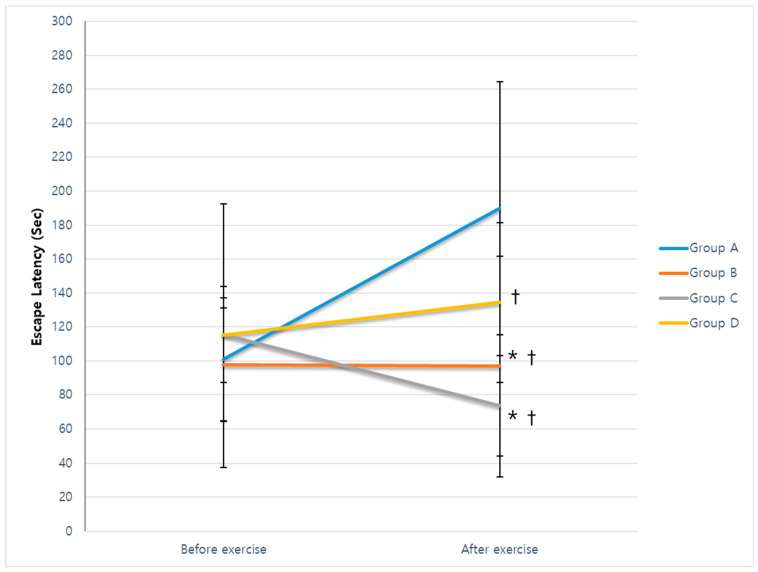
Escape latency during the Morris water maze test. No differences in escape latency on the day after cerebral infarction were found among the groups (*p* = 0.798). The follow-up escape latencies in groups B, C, and D were significantly shorter than those in group A, and latencies in groups B and C were shorter than those in group D (*p* = 0.001). Repeated-measures ANOVA with the Duncan post hoc test showed a significant time × group interaction (F = 5.851, *p* = 0.003). However, no significant effect was shown for time (F = 2.097, *p* = 0.157). *^,†^
*p* < 0.05.

**Figure 2 ijms-25-08997-f002:**
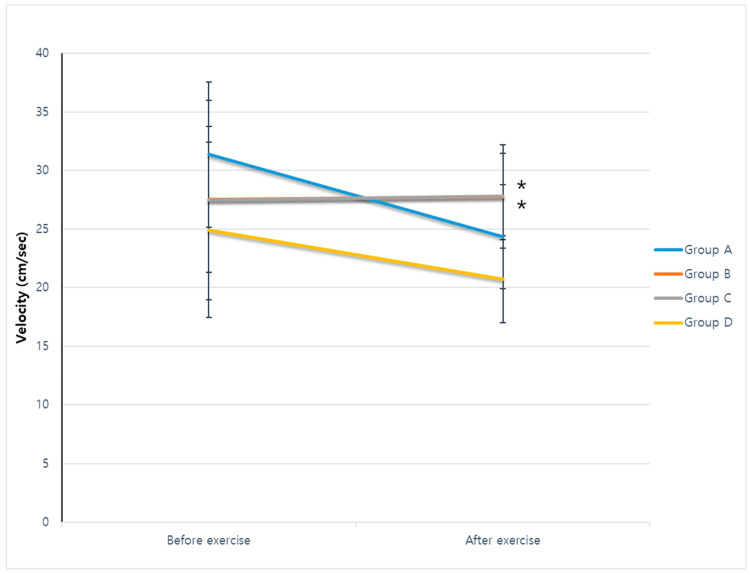
Velocity during the Morris water maze test. No differences in velocity were noted among groups on the day after cerebral infarction (*p* = 0.406). The Duncan post hoc correction revealed faster velocities in groups B and C compared with groups A and D after 4 weeks of exercise (*p* = 0.002). However, repeated-measures ANOVA with the Duncan post hoc test showed no significant effects for time and time × group interaction (F = 2.375, *p* = 0.133; F = 0.334, *p* = 0.567). * *p* < 0.05.

**Figure 3 ijms-25-08997-f003:**
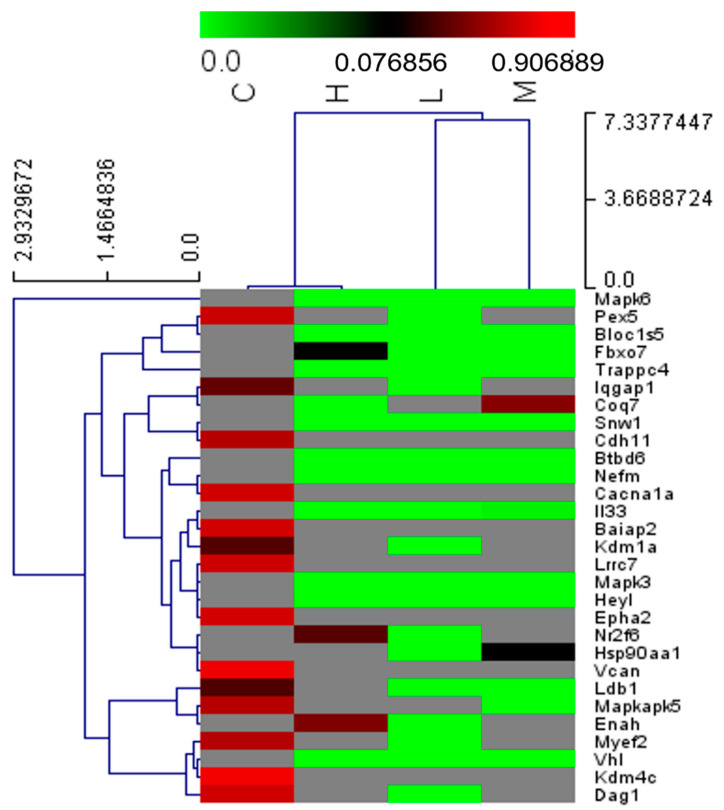
Hierarchical clustering heatmap in each group. Average normalized read count (RC) of gene expression and fold change of gene expression in each group compared with the control group (*p* < 0.05, fold change > 1.0, average normalized RC > 4). The patterns of gene expression in the low- and moderate-intensity treadmill exercise groups were similar and different from the control and high-intensity treadmill exercise groups. C: control group (no treadmill exercise); H: high-intensity treadmill exercise group; L: low-intensity treadmill exercise group; M: moderate-intensity treadmill exercise group.

**Figure 4 ijms-25-08997-f004:**
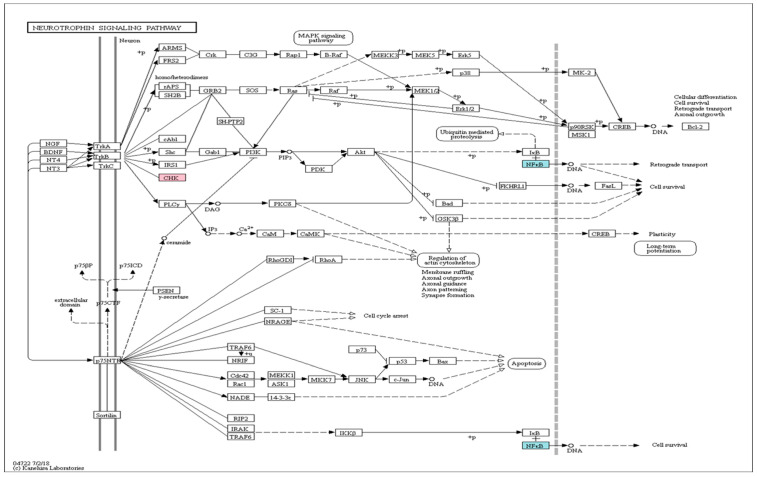
Neurotrophin-signaling pathways in the low-intensity exercise group (coral: upregulation; blue: downregulation). The *CHK* gene was upregulated, and the *NF-κB* gene was downregulated with low-intensity treadmill exercise. The neurotrophin-signaling pathway was elicited with the KEGG Mapper Search and Color Pathway tool (http://www.genome.jp/kegg/tool/map_pathway2.html, accessed on 19 December 2020).

**Figure 5 ijms-25-08997-f005:**
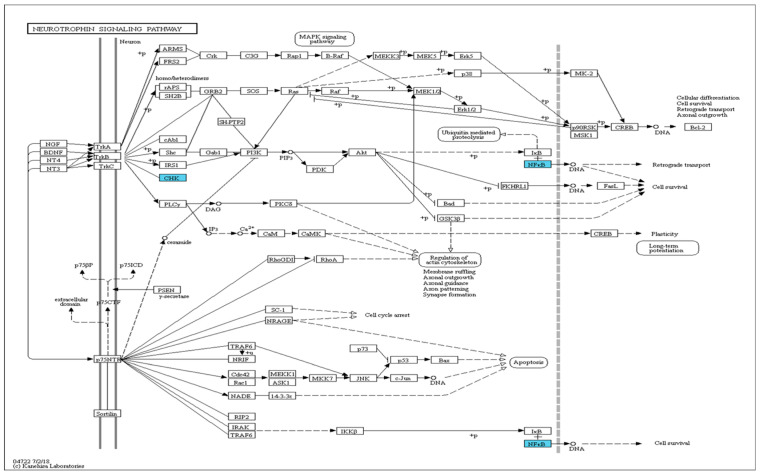
Neurotrophin-signaling pathways in the moderate-intensity exercise group (blue: downregulation). The *CHK* and *NF-κB* genes were both downregulated. The neurotrophin-signaling pathway was elicited with the KEGG Mapper Search and Color Pathway tool (http://www.genome.jp/kegg/tool/map_pathway2.html, accessed on 19 December 2020).

**Figure 6 ijms-25-08997-f006:**
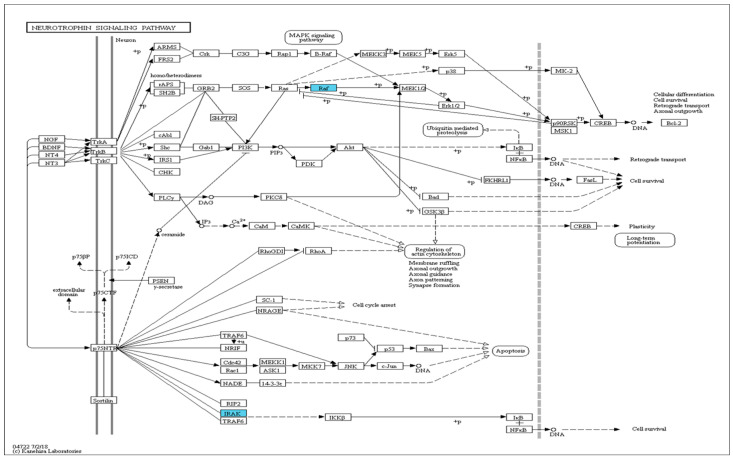
Neurotrophin-signaling pathways in the high-intensity exercise group (blue: downregulation). The *Raf* and *IRAK* genes were downregulated. The neurotrophin signaling pathway was elicited with the KEGG Mapper Search and Color Pathway tool (http://www.genome.jp/kegg/tool/map_pathway2.html, accessed on 19 December 2020).

**Figure 7 ijms-25-08997-f007:**
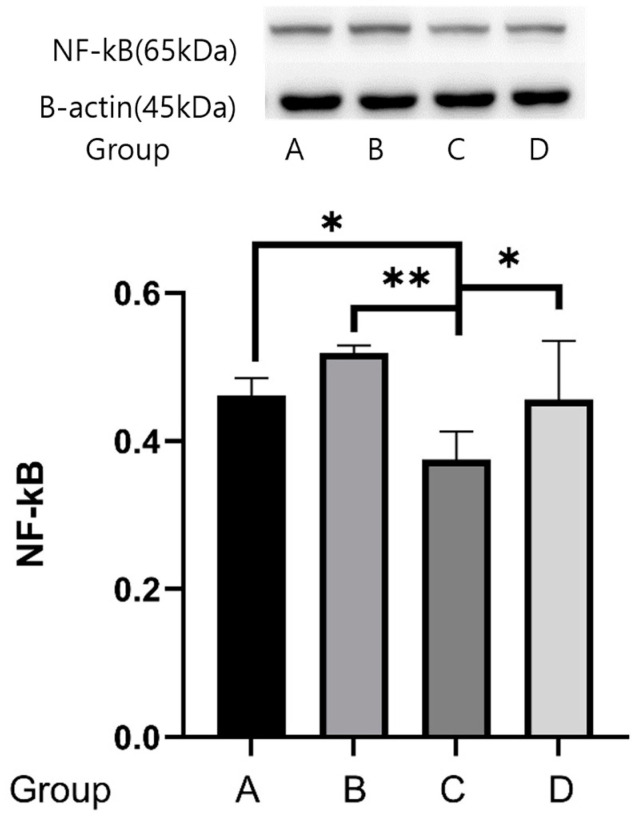
Effects of treadmill exercise on *NF-κB* protein concentrations. The reactivities of *NF-κB* protein were 0.461 ± 0.023 μg protein in group A, 0.518 ± 0.468 μg protein in group B, 0.375 ± 0.037 μg protein in group C, and 0.456 ± 0.079 μg protein in group D. The expression of the *NF-κB* protein in group C was weaker than in groups A, B, and D (* *p* = 0.05, ** *p* = 0.001).

**Figure 8 ijms-25-08997-f008:**
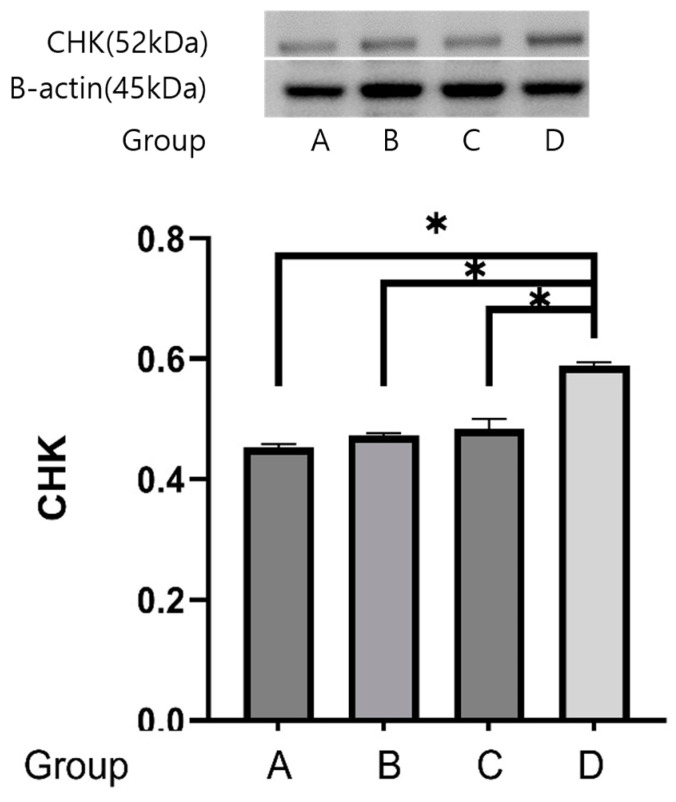
Effects of treadmill exercise on *CHK* protein concentrations. The reactivities of the *CHK* protein were 0.4532 ± 0.005 μg protein in group A, 0.4724 ± 0.003 μg protein in group B, 0.4835 ± 0.016 μg protein in group C, and 0.5883 ± 0.005 μg protein in group D. The expression of the *CHK* protein was stronger in group D than in groups A, B, and C. * *p* = 0.05.

**Figure 9 ijms-25-08997-f009:**
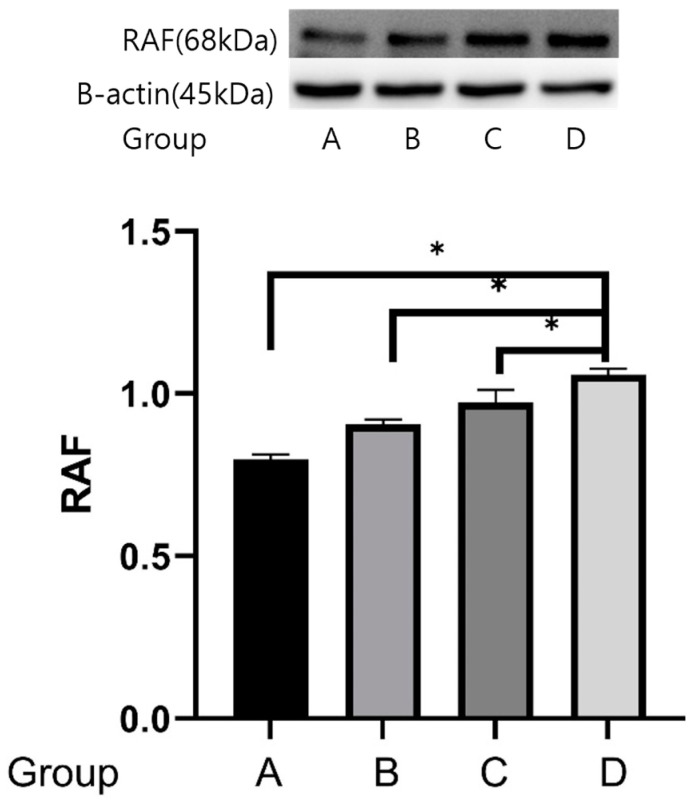
Effects of treadmill exercise on *Raf* protein concentrations. The reactivities of the *Raf* protein were 0.797 ± 0.016 μg protein in group A, 0.906 ± 0.014 μg protein in group B, 0.973 ± 0.038 μg protein in group C, and 1.059 ± 0.018 μg protein in group D. The expression of the *Raf* protein was stronger in group D than in groups A, B, and C. * *p* = 0.05.

**Figure 10 ijms-25-08997-f010:**
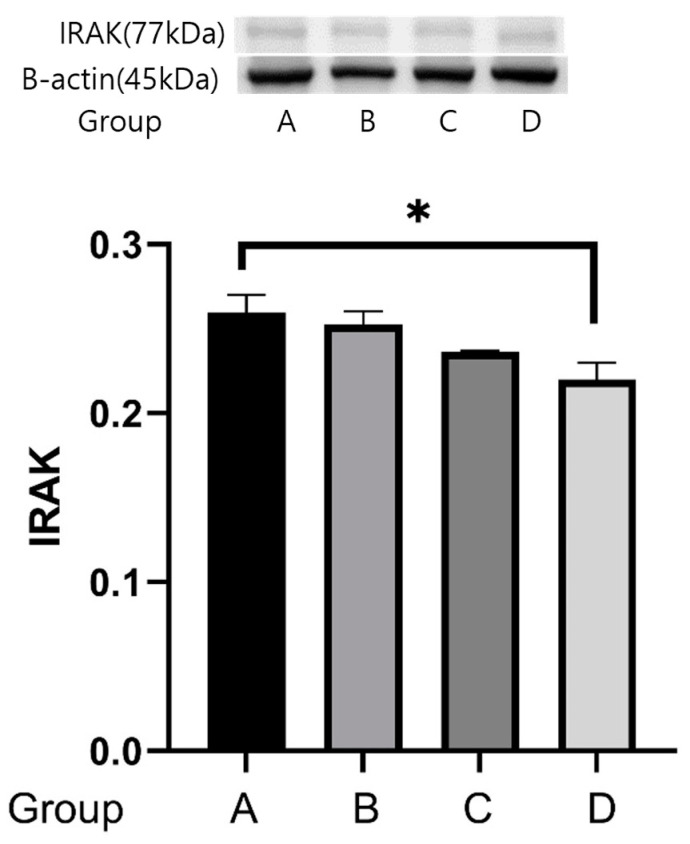
Effects of treadmill exercise on *IRAK* protein concentrations. The reactivities of *IRAK* protein were 0.252 ± 0.012 μg protein in group A, 0.247 ± 0.011 μg protein in group B, 0.236 ± 0.010 μg protein in group C, and 0.227 ± 0.014 μg protein in group D. The expression of the *IRAK* protein was weaker in group D than in group A. * *p* = 0.05.

**Figure 11 ijms-25-08997-f011:**
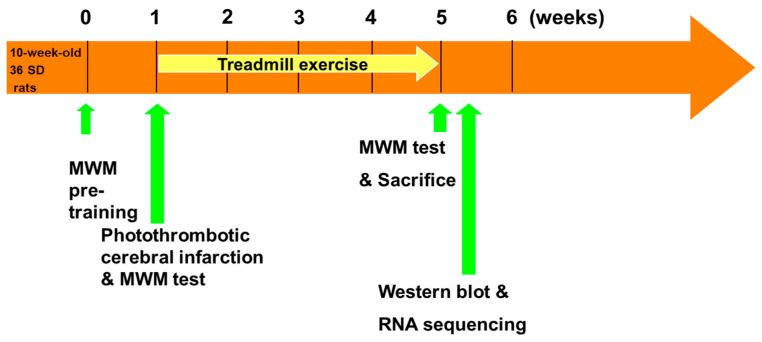
Schematic representation of the experiments. Photothrombotic cerebral infarction was conducted for 10-week-old male Sprague-Dawley (SD) rats (*n* = 36). All rats were acclimated to the abovementioned pre-training process for 3 consecutive days before photothrombotic cerebral infarction. The Morris water maze (MWM) test was performed before and after a regular treadmill exercise program (5 days/week, 4 weeks). The follow-up MWM test was performed after the treadmill exercise program; all rats were sacrificed after the MWM test to obtain the brain tissues.

## Data Availability

The data presented in this study are available on request from the corresponding author.

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
