# Peer review of "Gene Expression of Neurogenesis Related to Exercise Intensity in a Cerebral Infarction Rat Model"

_ijms, 2024, doi:10.3390/ijms25168997_

Round 1

Reviewer 1 Report

Comments and Suggestions for Authors

The study conducted by Song evaluated the therapeutic exercise intensity that would be most effective in improving cognitive functional recovery post stroke and associated this effect with genes related to neurogenesis. The study has an innovative proposal, but needs improvements for possible publication.

How in the abstract do the authors conclude with moderate intensity exercise if high intensity exercise also had effects on some genes?

There was a missing point after reference 15 on line 61;

It would be interesting for the authors to place the item “timing of experiments” before statistical analysis item, because in this place the authors have already read each methodology and better understood each experimental protocol;

Regardless of the age of the animals, it would be interesting for the animals to describe the experimental weeks starting from week 0 or 1, instead of 10;

In western blotting analyses, the phrase about membrane blockade is confusing. Was it blocked for 30 minutes or 1 hour?

It would be interesting for the authors to create the graphs in more specialized software, such as Prisma Graphpad or similar;

It is important that authors add details of the KEGG Mapper Tool Analysis in the methodology

In relation to the western blot assay, when evaluating NFKB expression, the group A b-actin band has a very low expression, which indicates a reduction in the protein in the sample and not in expression. Furthermore, in the CHK assessment, the bands in groups C and D do not appear to be higher compared to groups A and B, but rather behind the bands. Similar comment applies to the evaluation of RAF expression. When evaluating IRAK expression, there is no clear change in difference between the bands of groups A and D. Furthermore, as some of these proteins are phosphorylated, the interesting thing is to evaluate the phosphorylated protein by comparing its expression with its total protein;

To confirm the mechanisms involved in improving post-stroke cognition through aerobic exercise, it is interesting that the authors carry out direct analyzes such as immunofluorescence, ELISA, RT-PCR, in addition to improving the results with western blot.

Author Response

  1. Point by point response for reviewer

     Thank you for your helpful guidance and Reviewer’s informative comments for our manuscript.

    And we tried to solve the questions and comments more completely from the reviewer during the given period. We accomplished it and are submitting a revised version of manuscript ‘ijms-3107458’, entitled “Gene Expression of Neurogenesis related to Exercise Intensity in a Cerebral Infarction Rat Model” to be considered for publication in International Journal of molecular science. This manuscript has certainly benefited from these insightful comments and suggestions. We look forward to working with you and the reviewer to move this manuscript closer to publication in International Journal of Molecular Sciences.

    I have responded specifically to comments and suggestions as shown below. To make the changes easier to be identified, I have numbered them along each number from the reviewer’s questions and comments.

    =========================================

    1. How in the abstract do the authors conclude with moderate intensity exercise if high intensity exercise also had effects on some genes?

Answer: Thank you for your valuable comment. This study showed the changes in some genes in high intensity exercise. However, there was no improvement on behavioral test in high intensity exercise statistically. So, we finally conclude that moderate intensity exercise showed the effective exercise intensity on cognitive function through the behavioral test and western blot analysis 

  1. There was a missing point after reference 15 on line 61;

Answer: Thank you for your valuable comment. We add a missing point on line 61, just as below;

“acute high-intensity interval training may enhance the executive function in a previous systematically review [15]”

  1. It would be interesting for the authors to place the item “timing of experiments” before statistical analysis item, because in this place the authors have already read each methodology and better understood each experimental protocol;

Answer: Thank you for your valuable comment. We replace the item “timing of experiments” before statistical analysis as you recommended.

  1. Regardless of the age of the animals, it would be interesting for the animals to describe the experimental weeks starting from week 0 or 1, instead of 10;

Answer: Thank you for your valuable comment. We changed the figure “timing of experiments” as you recommended.

  1. In western blotting analyses, the phrase about membrane blockade is confusing. Was it blocked for 30 minutes or 1 hour?

Answer: Thank you for your valuable comment. We rewrote the methods about the blockage for 1 hour.

  1. It would be interesting for the authors to create the graphs in more specialized software, such as Prisma Graphpad or similar;

Answer: Thank you for your valuable comment. We create the graphs with Prisma Graphpad for western blot analysis.

  1. It is important that authors add details of the KEGG Mapper Tool Analysis in the methodology

Answer: Thank you for your valuable comment. We include the details of the KEGG Mapper tool analysis in the method section just as below;

“Neurotrophin signaling pathway was elicited by KEGG (Kyoto Encyclopedia of Genes and Genomes) mapper-Search & Color Pathway tool (http://www.genome.jp/kegg/tool/map_pathway2.html). We subjected the gene symbols with greater expression in the experimental groups than in the control group (p < 0.05, fold change > 1.0 times, average normalized RC > 4 times). The selected gene symbols were put the website and we clicked ‘included aliases’ and ‘uncolored diagram’ Then, the related pathways would be presented [53]. We chose the neurotrophin signaling pathway listed on the various pathways in each group.”

  1. In relation to the western blot assay, when evaluating NFKB expression, the group A b-actin band has a very low expression, which indicates a reduction in the protein in the sample and not in expression. Furthermore, in the CHK assessment, the bands in groups C and D do not appear to be higher compared to groups A and B, but rather behind the bands. Similar comment applies to the evaluation of RAF expression. When evaluating IRAK expression, there is no clear change in difference between the bands of groups A and D. Furthermore, as some of these proteins are phosphorylated, the interesting thing is to evaluate the phosphorylated protein by comparing its expression with its total protein;

Answer: Thank you for your valuable comment. We performed western blot analysis again and included new data with bands.

  1. To confirm the mechanisms involved in improving post-stroke cognition through aerobic exercise, it is interesting that the authors carry out direct analyzes such as immunofluorescence, ELISA, RT-PCR, in addition to improving the results with western blot.
    Answer: Thank you for your valuable comment. We include the limitation about the lack of other analysis such as immunofluorescence, ELISA, RT-PCR just as below;

“this study showed the gene expression with RNA sequencing analysis and performed the western blot analysis for quantification of protein related to gene expression. There are several research to use the quantitative methods as western blot analysis after RNA se-quencing [43-45]. However, there are several direct analyses such as immunofluorescence, ELISA or RT-PCR to confirm the mechanism. It would be more powerful to prove the quan-tify the gene expression with immunofluorescence, ELISA or RT-PCR. Further study would be needed to use direct analysis methods for standardizing the biomechanism of neuro-genesis after various exercise intensity.”

================================================

Thank you very much again.

Sincerely yours,

Sam-Gyu Lee, MD, PhD

Department of Physical & Rehabilitation Medicine,

Chonnam National University Medical School & Hospital,

#42, Jebong-Ro, Dong-Gu, Gwangju, 61469, Republic of Korea

Telephone: + 82 62 220 5180 (Gwangju office)

           +82 61 379 8282 (Hwasun office)

FAX: + 82 62 228 5975 (Gwangju office)

     +82 61 379 7779 (Hwasun office) 

       [email protected]

Reviewer 2 Report

Comments and Suggestions for Authors

We and others have shown that regular exercise enhance functional recovery including cognition, in stroke patients. However, the optimal intensity for promoting neurogenesis related to cognition remains unclear. In this study the authors investigated the best exercise intensity for functional recovery after stroke using RNA sequencing following regular treadmill exercise. To this end, they used 36 male Sprague–Dawley rats, 10 weeks old, with photothrombotic cerebral infarction. A Morris water maze (MWM) test was conducted before and after a 4-week treadmill exercise regimen (5 days/week). Rats were divided into four groups: A (no exercise), B (low-intensity, max speed 18 m/min), C (moderate-intensity, max speed 24 m/min), and D (high-intensity, max speed 30 m/min). After 4 weeks, the MWM test was repeated, rats were sacrificed, and RNA sequencing was performed on ipsilesional hippocampal tissue. They found no initial differences in escape latency and velocity among groups. After 4 weeks, groups B, C, and D had shorter escape latencies than group A, with groups B and C outperforming group D. Velocity was higher in groups A, B, and C compared to group D. RNA sequencing identified 30 genes related to neurogenesis. In the neurotrophin signaling pathway, low-intensity exercise upregulated CHK and downregulated NF-κB. Moderate-intensity exercise downregulated both CHK and NF-κB. High-intensity exercise downregulated Raf and IRAK. Western blot analysis showed the lowest NF-κB in moderate-intensity, elevated CHK and Raf, and further decreased IRAK in high-intensity.

They concluded that moderate-intensity exercise may enhance neuroplasticity. Differences in neurotrophin expression related to exercise intensity could explain the mechanisms of neuroplasticity, highlighting NF-κB as a key factor in exercise-induced neurogenesis.

The study is of interest, but it does not reflect aging and comorbidities as the most important risk factors for stroke (see, DOI: 10.1007/s11357-021-00483-2; DOI: 10.1111/ACEL.12678)This is a severe study limitation that must be mentioned in the Discussion section.

Author Response

Point by point response for reviewer

 Thank you for your helpful guidance and Reviewer’s informative comments for our manuscript.

And we tried to solve the questions and comments more completely from the reviewer during the given period. We accomplished it and are submitting a revised version of manuscript ‘ijms-3107458’, entitled “Gene Expression of Neurogenesis related to Exercise Intensity in a Cerebral Infarction Rat Model” to be considered for publication in International Journal of molecular science. This manuscript has certainly benefited from these insightful comments and suggestions. We look forward to working with you and the reviewer to move this manuscript closer to publication in BMC neuroscience.

I have responded specifically to comments and suggestions as shown below. To make the changes easier to be identified, I have numbered them along each number from the reviewer’s questions and comments.

=========================================

The study is of interest, but it does not reflect aging and comorbidities as the most important risk factors for stroke (see, DOI: 10.1007/s11357-021-00483-2; DOI: 10.1111/ACEL.12678). This is a severe study limitation that must be mentioned in the Discussion section.

Answer: Thank you for your valuable comment. We include the limitation about the lack of consideration for aging and comorbidities as the most important risk factors for stroke.

================================================

Thank you very much again.

Sincerely yours,

Sam-Gyu Lee, MD, PhD

Department of Physical & Rehabilitation Medicine,

Chonnam National University Medical School & Hospital,

#42, Jebong-Ro, Dong-Gu, Gwangju, 61469, Republic of Korea

Telephone: + 82 62 220 5180 (Gwangju office)

           +82 61 379 8282 (Hwasun office)

FAX: + 82 62 228 5975 (Gwangju office)

     +82 61 379 7779 (Hwasun office) 

       [email protected]

Round 2

Reviewer 1 Report

Comments and Suggestions for Authors

The authors fulfilled most of the requests to improve the manuscript, making it eligible for acceptance.

Author Response

Point by point response for reviewer

 Thank you for your helpful guidance and Reviewer’s informative comments for our manuscript.

And we tried to solve the questions and comments more completely from the reviewer during the given period. We accomplished it and are submitting a revised version of manuscript ‘ijms-3107458’, entitled “Gene Expression of Neurogenesis related to Exercise Intensity in a Cerebral Infarction Rat Model” to be considered for publication in International Journal of molecular science. This manuscript has certainly benefited from these insightful comments and suggestions. We look forward to working with you and the reviewer to move this manuscript closer to publication in International Journal of Molecular Sciences.

Thank you for your precise reviews. 

Sincerely yours,

Sam-Gyu Lee, MD, PhD

Department of Physical & Rehabilitation Medicine,

Chonnam National University Medical School & Hospital,

#42, Jebong-Ro, Dong-Gu, Gwangju, 61469, Republic of Korea

Telephone: + 82 62 220 5180 (Gwangju office)

           +82 61 379 8282 (Hwasun office)

FAX: + 82 62 228 5975 (Gwangju office)

     +82 61 379 7779 (Hwasun office) 

       [email protected]

Reviewer 2 Report

Comments and Suggestions for Authors

The recommendation was to highlight that advanced age and co-morbidities are severe limiting factors that impede post-stroke recovery, as well as to note the caveats of the model employed. The sentence added by the authors reflects the factor of aging but not the factor of co-morbidities: "Previous studies reported that aged 267 rats could limit the recovery of neurological function due to the lack of neuroprotective factors [41,42]." Therefore, the appropriate references are DOI: 10.1007/s11357-021-00483-2 and DOI: 10.1111/ACEL.12678, which should replace the older references that were cited.

Comments on the Quality of English Language

English needs improvement

Author Response

Point by point response for reviewer

 Thank you for your helpful guidance and Reviewer’s informative comments for our manuscript.

And we tried to solve the questions and comments more completely from the reviewer during the given period. We accomplished it and are submitting a revised version of manuscript ‘ijms-3107458’, entitled “Gene Expression of Neurogenesis related to Exercise Intensity in a Cerebral Infarction Rat Model” to be considered for publication in International Journal of molecular science. This manuscript has certainly benefited from these insightful comments and suggestions. We look forward to working with you and the reviewer to move this manuscript closer to publication in International Journal of Molecular Sciences.

I have responded specifically to comments and suggestions as shown below. To make the changes easier to be identified, I have numbered them along each number from the reviewer’s questions and comments.

=========================================

  1. The recommendation was to highlight that advanced age and co-morbidities are severe limiting factors that impede post-stroke recovery, as well as to note the caveats of the model employed. The sentence added by the authors reflects the factor of aging but not the factor of co-morbidities: "Previous studies reported that aged 267 rats could limit the recovery of neurological function due to the lack of neuroprotective factors [41,42]." Therefore, the appropriate references are DOI: 10.1007/s11357-021-00483-2 and DOI: 10.1111/ACEL.12678, which should replace the older references that were cited.

Answer: Thank you for your valuable comment. We include the limitation about the lack of consideration for aging and comorbidities as you recommended.

Answer: Thank you for your valuable comment. We include the limitation about the lack of consideration for aging and comorbidities as you recommended.

“Previous studies reported that aged rats could have limited recovery of neurologic function because of the lack of neuroprotective factors, immune imbalance, and comorbidities related to stroke risk factors such as obesity and metabolic syndrome. [41-43]. Further studies are needed to consider aging and comorbidities in stroke experiments.”

  1. Comments on the Quality of English Language : English needs improvement

Answer: Thank you for your comment. I did the English editing service and included the certification of English editing.

================================================

Thank you very much again.

Sincerely yours,

Sam-Gyu Lee, MD, PhD

Department of Physical & Rehabilitation Medicine,

Chonnam National University Medical School & Hospital,

#42, Jebong-Ro, Dong-Gu, Gwangju, 61469, Republic of Korea

Telephone: + 82 62 220 5180 (Gwangju office)

           +82 61 379 8282 (Hwasun office)

FAX: + 82 62 228 5975 (Gwangju office)

     +82 61 379 7779 (Hwasun office) 

       [email protected]

Round 3

Reviewer 2 Report

Comments and Suggestions for Authors

The authors have adequately addressed my concerns. The manuscript can be published in its current form

Comments on the Quality of English Language

Good for publication purposes